# Improve Training Stability of Semi-supervised Generative Adversarial Networks with Collaborative Training

## Abstract

Improved generative adversarial network (Improved GAN) is a successful method of using generative adversarial models to solve the problem of semi-supervised learning. However, it suffers from the problem of unstable training. In this paper, we found that the instability is mostly due to the vanishing gradients on the generator. To remedy this issue, we propose a new method to use collaborative training to improve the stability of semi-supervised GAN with the combination of Wasserstein GAN. The experiments have shown that our proposed method is more stable than the original Improved GAN and achieves comparable classification accuracy on different data sets.

## 1 Introduction

Generative adversarial networks (GANs) (Goodfellow et al., 2015) have been recently studied intensively and achieved great success in deep learning domain(Salimans et al., 2017; Mao et al., 2016; Springenberg, 2015; Arjovsky & Bottou, 2017; Arjovsky et al., 2017). A typical GAN simulates a two-player minimax game, where one aims to fool the other and the overall system is finally able to achieve equilibrium.

Specifically speaking, we have a generator $G$ to generate fake data $G(z)$ from a random variable $z$ whose distribution density is $p(z)$, and also we have a discriminator $D(x)$ to discriminate the real $x$ from the generated data $G(z)$, where $x \sim p_r(x)$ and $p_r$ is the distribution density of real data. We optimize the two players $G(z)$ and $D(x)$ by solving the following minimax problem:

$$
\begin{aligned}
G^*, D^* = \arg\min_G \max_D \mathbb{E}_{x \sim p_r(x)} \left\{ \log(D(x)) \right\} \\
+ \mathbb{E}_{z \sim p(z)} \left\{ \log(1 - D(G(z))) \right\}.
\end{aligned}
\tag{1}
$$

This method is so called as the original GAN (Goodfellow et al., 2015). After this, many different types of GANs have been proposed, e.g., least-squared GAN (Mao et al., 2016), cat-GAN (Springenberg, 2015), W-GAN (Arjovsky & Bottou, 2017; Arjovsky et al., 2017), Improved GAN (Salimans et al., 2017), so on and so forth, focusing on improving the performance of GANs and extending the GAN idea to other application scenarios.

For instance, the original GAN is trained in a completely unsupervised learning way (Goodfellow et al., 2015), along with many variants, such as LS-GAN and cat-GAN. It was later extended to semi-supervised learning. In (Salimans et al., 2017), Salimans et al. proposed the Improved GAN to enable generation and classification of data simultaneously. In (Li et al., 2017), Li et al. extended this method to consider conditional data generation.

Another issue regarding the unsupervised learning of GANs is the lack of training stability in the original GANs, mostly because of dimension mismatch (Arjovsky & Bottou, 2017). A lot of efforts have been dedicated to solve this issue. For instance, in (Arjovsky & Bottou, 2017; Arjovsky et al., 2017), the authors theoretically found that the instability problem and dimension mismatch of the unsupervised learning GAN was due to the maxing out of Jensen-Shannon divergence between the true and fake distribution and therefore proposed using the Wasserstein distance to train GAN. However, to calculate the Wasserstein distance, the network functions are required to be 1-Lipschitz,

which was simply implemented by clipping the weights of the networks in (Arjovsky et al., 2017). Later, Gulrajani et. al. improved it by using gradient penalty (Gulrajani et al., 2017). Besides them, the same issue was also addressed from different perspectives. In (Roth et al., 2017), Roth et al. used gradient norm-based regularization to smooth the f-divergence objective function so as to reduce dimension mismatch. However, the method could not directly work on f-divergence, which was intractable to solve, but they instead optimized its variational lower bound. Its converging rate is still an open question and its computational complexity may be high. On the other hand, there were also some efforts to solve the issue of mode collapse, so as to try to stabilize the training of GANs from another perspective, including the unrolled method in (Metz et al., 2016), mode regularization with VAEGAN (Che et al., 2016), and variance regularization with bi-modal Gaussian distributions (Karan Grewal, 2017). However, all these methods were investigated in the context of unsupervised learning. Instability issue for semi-supervised GAN is still open.

In this work, we focus on investigating the training stability issue for semi-supervised GAN. To the authors' best knowledge, it is the first work to investigate the training instability for semi-supervised GANs, though some were done for unsupervised GANs as aforementioned. The instability issue of the semi-supervised GAN (Salimans et al., 2017) is first identified and analyzed from a theoretical perspective. We prove that this issue is in fact caused by the vanishing gradients theorem on the generator. We thus propose to solve this issue by using collaborative training to improve its training stability. We theoretically show that the proposed method does not have vanishing gradients on the generator, such that its training stability is improved. Besides the theoretical contribution, we also show by experiments that the proposed method can indeed improve the training stability of the Improved GAN, and at the same time achieve comparable classification accuracy.

It is also worth to note that (Li et al., 2017) proposed the Triple GAN that also possessed two discriminators. However, its purpose is focused on using *conditional* probability training (the original GAN uses unconditional probability) based on data labels to improve the training of GAN, but not on solving the instability issue. Therefore, the question of instability for the Triple GAN is still unclear. More importantly, the method, collaborative training, proposed for exploring the data labels with only unconditional probability in this paper , can also be applied to the Triple GAN to improve its training stability, in the case of conditional probability case.

The rest of the paper is organized as follows: in Section 2, we present the generator vanishing gradient theorem of the Improved GAN. In Section 3, we propose a new method, collaborative training Wasserstein GAN (CTW-GAN) and prove its nonvanishing gradient theorem. In Section 4, we present our experimental results and finally give our conclusion in Section 5.

## 2 TRAINING INSTABILITY OF IMPROVED GAN

The improved GAN (Salimans et al., 2017) combines supervised and unsupervised learning to solve the semi-supervised classification problem by simulating a two-player minmax game with adversarial training. The adversarial training is split into two steps. In the first step, it minimizes the following objective function for discriminator $D$ for data $x$ and labels $y$:

$$
\begin{aligned}
D^* = \arg \min_D \; & \mathbb{E}_{(x,y)\sim p_r(x,y)} \|D(x) - y\|_2^2 \\
& + \mathbb{E}_{x\sim p_r(x), z\sim p_g(z)} \left[ D(x) - D(G(z)) \right].
\end{aligned}
\tag{2}
$$

In the second step, it minimizes the distance of feature matching to optimize the generator $G$:

$$
G^* = \arg \min_G \mathcal{L}_g(G)
\tag{3}
$$

where

$$
\mathcal{L}_g(G) = \mathbb{E}_{z\sim p(z),x\ p_r(x)} \|D^{(-3)}(G(z)) - D^{(-3)}(x)\|_2^2,
\tag{4}
$$

and $D^{(-3)}(x)$ are the outputs from the $(n-3)$-th layer for a net with $n$ layers.

### 2.1 GENERATOR VANISHING GRADIENT THEOREM FOR IMPROVED GAN

In this subsection, we prove the theorem of vanishing gradients on the generator for Improved GAN. This explains why the Improved GAN lacks training stability, as showed on some datasets, such as MNIST (cf. Section 4).

**Theorem 2.1 (Vanishing gradients on the generator for Improved GAN)** *Let $g_\theta : \mathcal{Z} \to \mathcal{X}$ be a differentiable function that induces a distribution $\mathbb{P}_g$. Let $\mathbb{P}_r$ be the real data distribution. Let $D$ be a differentiable discriminator bounded by $\mathcal{T}$, i.e., $\|D(x)\|_2 \leq \mathcal{T}$. If the discriminator is trained to converge, i.e., $\|D - D^*\|_2 < \epsilon$, and $\mathbb{E}_{z \sim p(z)}[\|J_\theta g_\theta(z)\|_2^2] \leq \mathcal{M}^2$, then*

$$\|\nabla_\theta \mathbb{E}\left(D_{x \sim p_r}(x) - D_{z \sim p(z)}(g_\theta(z))\right)^2\|_2^2$$
$$\leq 8(\mathcal{T} + \epsilon)^2 \cdot \mathcal{M}^2 \cdot \epsilon^2 \tag{5}$$
$$\to 0.$$

**Proof 2.1** *See Appendix A.1.*

This theorem implies that the generator gradients vanish when the discriminator is trained to converge. In this case, the generator training saturates, which explains the training instability phenomenon of the Improved GAN. The way to solve this problem is our next question.

## 3 COLLABORATIVE TRAINING OF IMPROVED GAN WITH WASSERSTEIN GAN (CTW-GAN)

In this section, we propose a new method to solve the instability issue of the Improved GAN by using collaborative training between two GANs. These two GANs contribute to the adversarial training from two different perspectives, which may help avoid the drawbacks of each one. This is the basic idea behind the proposed method. The detailed procedure of CTW-GAN can be summarized as a minimax game carried out in two steps:

At the first step, the discriminators $D_c$ and $D_w$ are optimized simultaneously:

$$D_c^*, D_w^* = \arg \min_{D_c, D_w, \|D_w\|_L \leq 1} \mathcal{L}(D_c, D_w), \tag{6}$$

where $\|D_w\|_L \leq 1$ means $D_w$ is 1-Lipschitz and

$$\mathcal{L}(D_c, D_w) = \mathbb{E}_{(x,y) \sim p_r(x,y)} \|D_c(x) - y\|_2^2$$
$$+ E_{x \sim p_r(x), z \sim p_g(z)} \{D_c(x) - D_c(G(z))\} \tag{7}$$
$$+ \mathbb{E}_{z \sim p_g(z)} \{D_w(G(z))\} - \mathbb{E}_{x \sim p_r(x)} \{D_w(x)\}$$

At the second step, the generator $G$ is then optimized by applying the optimized two discriminators $D_c, D_w$ to $G$:

$$G^* = \arg \min_G \mathcal{L}_g(G), \tag{8}$$

where

$$\mathcal{L}_g(G) = \lambda \cdot E_{z \sim p(z), x \sim p_r(x)} \|D_c^{(-3)}(G(z)) - D_c^{(-3)}(x)\|_2^2$$
$$- (1 - \lambda) \cdot E_{z \sim p_g(z)} D_w(G(z)). \tag{9}$$

The overall architecture for CTW-GAN is described by Figure 1, where $x_r$ and $x_u$ stand for labeled and unlabeled data respectively:

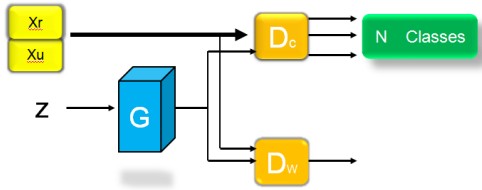

Figure 1: Architecture for CTW-GAN

## 3.1 Nonvanishing gradients theorem for CTW-GAN

Bearing in mind the generator vanishing gradients theorem for Improved GAN, we may ask if a similar problem exists for our proposed CTW-GAN. In the following, we prove that our proposed method does not have the vanishing gradients issue on the generator, which therefore improves the training stability of the original Improved GAN.

**Theorem 3.1** *Let $\mathbb{P}_r$ be any distribution. Let $\mathbb{P}_\theta$ be the distribution of $g_\theta(z)$ with $z$ being a random variable with a density $p$ and $g_\theta$ a continuous function with respect to $\theta$. Then there is a set of solutions $D_c, D_w$ to the problem*

$$\min_{D_c, D_w, \|D_w\|_L \leq 1} \mathcal{L}(D_c, D_w) \tag{10}$$

*and we have*

$$\nabla_\theta \mathcal{L}_g = (\lambda - 1)\mathbb{E}_{z \sim p(z)}[\nabla_\theta D_w(g_\theta(z))]. \tag{11}$$

*where the last term is the gradients of the Wasserstein distance $W(\mathbb{P}_r, \mathbb{P}_\theta)$, i.e.,*

$$\mathbb{E}_{z \sim p(z)}[\nabla_\theta D_w(g_\theta(z))] = \nabla_\theta W(\mathbb{P}_r, \mathbb{P}_\theta) \tag{12}$$

*when the term $\mathcal{L}(D_c, D_w)$ is well-defined.*

**Proof 3.1** *see Appendix A.2.*

*Remark*: the above $\|D_w\|_L \leq 1$ is required to be 1-Lipschitz. The constraint can be realized by weight clipping (Arjovsky et al., 2017) or gradient penalty (Gulrajani et al., 2017).

The proposed algorithm is described as follows:

---
**Algorithm 1** CTW-GAN with gradient penalty:

---
**Require:** Gradient penalty $\lambda_p = 10$, generator weight$\lambda_g$, and Adam hyperparameter $\alpha = 0.0001$
**Require:** Initial parameter $\theta_w$ for $D_w$, $\theta_c$ for $D_c$ and $\theta_g$ for $G$.
  **while** $\theta_g$ does not converge **do**
    **for** $i = 1 \cdots m$ **do**
      Sample a real data $x \sim p_r$ and a noisy data $z \sim p(z)$
      $L_c^{(i)} \leftarrow \|D_c(x) - y\|_2^2 + (D_c(x) - D_c(G(z)))$
    **end for**
    $\theta_c \leftarrow Adam(\nabla_{\theta_c} \frac{1}{m} \sum_{i=1}^m L_c^{(i)}, \alpha)$
    **for** $t = 1 \dots n_{critic}$ **do**
      **for** $i = 1 \cdots m$ **do**
        Sample a real data $x \sim p_r$, a noisy data $z \sim p(z)$ and a random variable $\epsilon \sim U(0, 1)$
        $\tilde{x} \leftarrow G_{\theta_g}(z)$
        $\hat{x} \leftarrow \epsilon x + (1 - \epsilon)\tilde{x}$
        $L_w^{(i)} \leftarrow (D_w(x) - D_w(G(z))) + \lambda_p(\|\nabla_{\hat{x}}(D_w(\hat{x}))\|_2 - 1)^2$
      **end for**
      $\theta_w \leftarrow Adam(\nabla_{\theta_c} \frac{1}{m} \sum_{i=1}^m L_c^{(i)}, \alpha)$
      sample a batch of latent variables $\{z^{(i)}\}_{i=1}^m \sim p(z)$
      $\theta_g \leftarrow Adam(\nabla_{\theta_g}\{\lambda_g \frac{1}{m} \sum_{i=1}^m (D_c(x) - D_c(G(z))^2 - (1 - \lambda_g)\frac{1}{m} \sum_{i=1}^m D_w(G(z))\}, \alpha)$
    **end for**
  **end while**

---

## 4 Experiments

In this section, we shall present the experiments to evaluate the proposed method. Our evaluation goals are twofold. On one hand, we evaluate the stability of CTW-GAN in comparison to that of the original Improved GAN to see whether our proposed method improves the training stability or not. On the other hand, we evaluate whether the proposed method achieves comparable classification performance to the original Improved GAN. To this end, we run experiments on two datasets: MNIST and CIFAR-10.

## 4.1 Experiments on MNIST

MNIST includes 50, 000 training images, 10, 000 validation images and 10, 000 testing images, which contain handwritten digits with size $28 \times 28$.

Following (Salimans et al., 2017), we randomly select a small set of labeled data from the 60, 000 training and validation set to perform semi-supervised learning with the selection size of 20, 50, 100, and 200 labeled examples. We run our experiments 9 times by giving the program different seeds. We use the seeds from $1 - 9$. For each seed, the labeled data is selected so as to have a balanced number of examples from each class. The rest of the training images are used as unlabeled data.

In our method, we use three networks whose architectures are described in Figure 2. We use batch normalization and add Gaussian noise to the output of each layer of the two discriminators as the original Improved GAN does (Salimans et al., 2017).

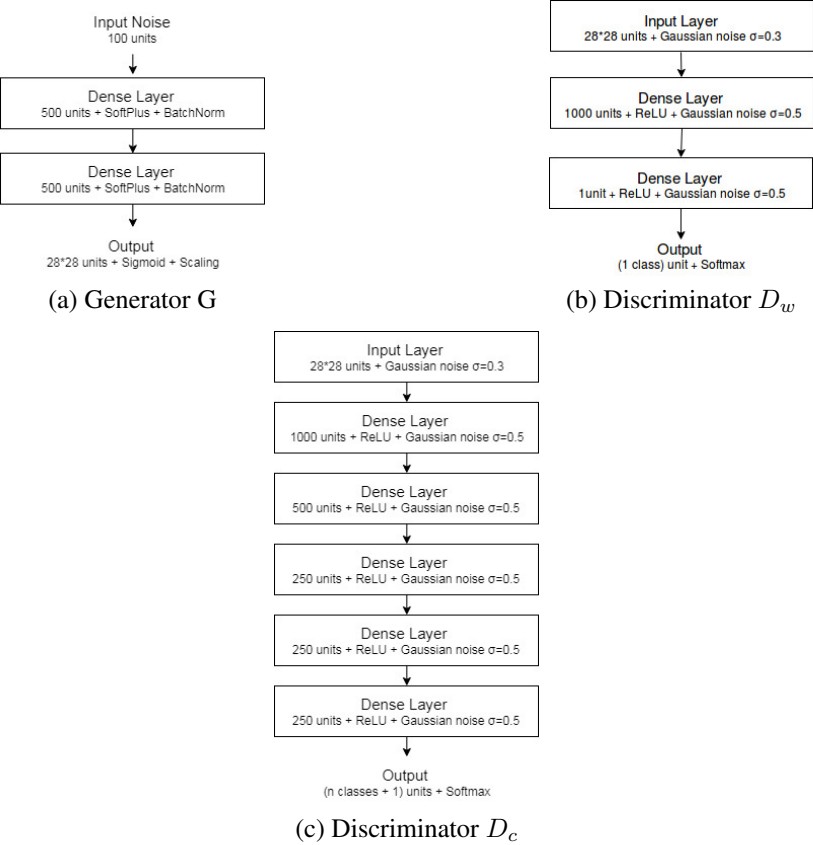

(a) Generator G

(b) Discriminator $D_w$

(c) Discriminator $D_c$

Figure 2: Network architectures used for MNIST

We only tune the parameter $\lambda = 0.1, 0.5$ from two values on the MNIST dataset. We do not tune any other parameters, such as learning rate, step size, etc.: we keep these as in the original Improved GAN. The results shown in Table 1 are reported with $\lambda = 0.1$, the threshold for gradient penalty is 10 and $n_{critic} = 5$:

From the results, we can easily see that the original improved GAN has one or two out of nine runs for training failure (unexpected high error rates and poor generate image quality). However, for our proposed method, no training failure occurs. This shows that our method improves the training stability indeed. On the other hand, besides making the training process more stable, our proposed method does not reduce the classification accuracy at all, which is beyond our original purpose of avoiding training instability of the Improved GAN. Reasoning it, it may imply that the information explored by the two discriminators may be very different, thus reflecting a distinct

| Method | n=50 | n=100 | n=200 |
|---|---|---|---|
| DGN (Kingma et al., 2014) | | 3.33($\pm$0.14) | |
| Virtual Adversarial (Miyato et al., 2015) | | 2.12 | |
| Cat-GAN (Springenberg, 2015) | | 1.91 ($\pm$0.10) | |
| Skip Deep Generative Model (Maaløe et al., 2016) | | 1.32 ($\pm$0.07) | |
| Ladder network (Rasmus et al., 2015) | | 1.06 ($\pm$0.37) | |
| Auxiliary Deep Generative Model (Maaløe et al., 2016) | | 0.96 ($\pm$0.02) | |
| Improved-GAN (including failure cases) | 6.46 ($\pm$6.95)(1F[1]) | 3.73($\pm$5.62)(2F) | 1.96($\pm$3.11)(1F) |
| Improved-GAN (only success cases) | 4.15 ($\pm$2.49) | 1.01($\pm$0.31) | 0.92($\pm$0.13) |
| Ours | **2.47**($\pm$1.37)**(0F)** | **0.85**($\pm$0.12)**(0F)** | **0.80**($\pm$0.05)**(0F)** |

Table 1: Number of incorrectly classified test examples for the semi-supervised setting on permutation invariant MNIST. Results are averaged over 9 seeds. Here "$n$F" means the number of training failure, i.e., instable training, occur during the training of Improved GAN.

| Method | Error rates |
|---|---|
| Cat-GAN (Springenberg, 2015) | 20.40 ($\pm$0.47) |
| Ladder network (Rasmus et al., 2015) | 19.58 ($\pm$0.46) |
| Improved-GAN | 0.1726 ($\pm$0.0032) |
| Ours | **0.1713** ($\pm$**0.0014**) |

Table 2: Test errors on semi-supervised CIFAR-10. Results are averaged over three splits of data. There is no failure case found in three runs for the original GAN on CIFAR-10. We use $4000$ labeled samples.

source of information for data representation. Utilizing those different information sources may help to improve classification accuracy, as long as the source of information is meaningful to some extent, or at least not noise. In our method, we use a very simple network for $D_w$ with only two layers. It may be possible to further improve classification performance if a network with more layers is used. We leave it for future work.

## 4.2 EXPERIMENTS ON CIFAR-10

In this section, we test our proposed method on the data set of CIFAR-10. CIFAR-10 consists of colored images belonging to 10 classes: airplane, automobile, bird, cat, deer, dog, frog, horse, ship and truck. There are 50,000 training and 10,000 testing samples with the size of $32 \times 32$. We split 5,000 training data of CIFAR-10 for validation if needed. Following (Salimans et al., 2017), we use a 9 layer deep convolutional network with dropout and weight normalization for the discriminator $D_c$. The generator $G$ is a 4 layer deep CNN with batch normalization. We use a very simple network with three layers for the discriminator $D_w$, due to the limiting GPU resource. The network architectures are given in Figure 3.

Table 2 summarizes our results on the semi-supervised learning task.

On CIFAR-10 dataset, it is interesting to see that there is no failure case found for the Improved GAN in three runs at the moment. From the theoretical viewpoint, this may be due to the abundant richness of the image features in color being much harder to be modeled by the neural nets than that of MNIST in grayscale. Thus, the discriminator $D_c$ trained on CIFAR-10 does not as easily converge as the one trained on MNIST, such that the gradients on the generator do not vanish. However, it does not mean that this possibility is avoided. In certain cases, as long as the discriminator is trained to converge., e.g., running more iterations than the generator, the gradients on the generator will surely vanish, as theoretically guaranteed by Theorem 2.1. On the other hand, our proposed method is still able to achieve comparable results to the original Improved GAN, besides providing a theoretical guarantee to the training stability. Due to the limiting GPU resource, we use a very simple network for $D_w$. In this sense, the characteristics captured by this network may not be rich enough. However, the results showed that even with the very simple network, the classification performance obtained

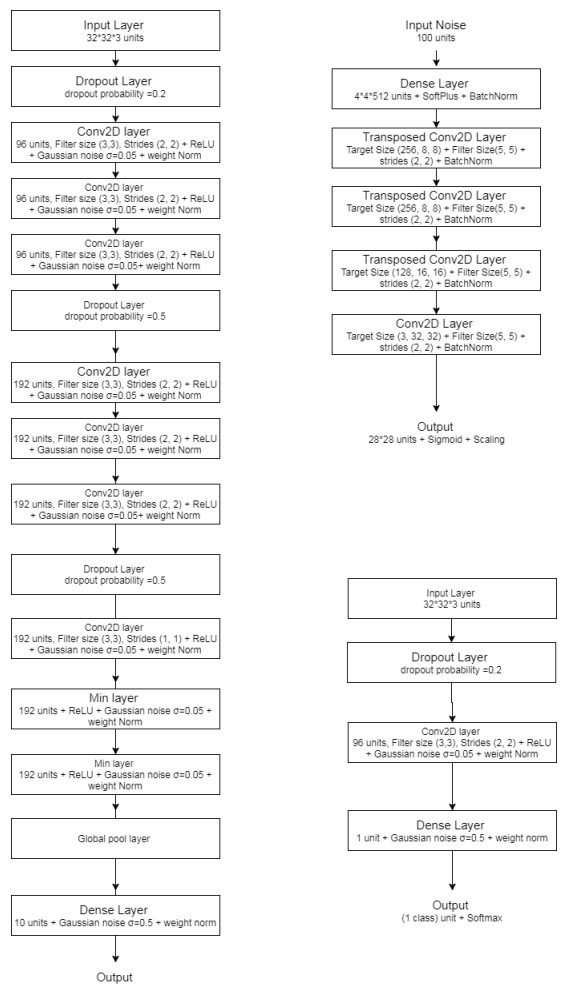

Figure 3: Network architectures used for CIFAR-10: The left net ($D_c$); the top right ($G$); the bottom right ($D_w$).

is roughly comparable to that of the Improved GAN. We expect that it would be possibly improved further if we have more GPU resources and are able to train a deeper network for $D_w$.

## 5 CONCLUSION

In the paper, we study the training instability issue of semi-supervised improved GAN. We have found that the training instability is mainly due to the vanishing gradients on the generator of the Improved GAN. In order to make the training of the Improved GAN more stable, we propose a collaborative training method to combine Wasserstein GAN with the semi-supervised improved GAN. Both theoretical analysis and experimental results on MNIST and CIFAR-10 have shown the effectiveness of the proposed method to improve training stability of the Improved GAN. In addition, it also achieves the classification accuracy comparable to the original Improved GAN.

ACKNOWLEDGMENTS

We would like to thank National Natural Science Foundation of China (61471205) for previously supporting the authors to prepare for the knowledge and skills demanded by this work.

## A    APPENDIX

### A.1    PROOF OF THEOREM 2.1

We only want the differentiation to the parameter $\theta$. For this, we can throw away the first term and only focus on the second term.

$$
\begin{aligned}
&\| \bigtriangledown \mathbb{E} \left( D_{x \sim p_r}(x) - D_{z \sim p(z)}(g_\theta(z)) \right)^2 \|_2^2 \\
&\leq \mathbb{E} \left\{ 2\| D_{x \sim p_r(x)}(x) - D_{z \sim p(z)}(g_\theta(z)) \|_2^2 \cdot \| \bigtriangledown_\theta D[g_\theta(z)] \|_2^2 \right\} \\
&\leq \mathbb{E} \left\{ 2\| D_{x \sim p_r(x)}(x) - D_{z \sim p(z)}(g_\theta(z)) \|_2^2 \cdot \| \bigtriangledown_x D[g_\theta(z)] \|_2^2 \right. \\
&\left. \quad \cdot \| J_\theta g_\theta(z) \|_2^2 \right\} \\
&\leq 2 \cdot \mathbb{E} \left\{ [ \| D_{x \sim p_r(x)}^*(x) + \epsilon \cdot I \|_2 + \| D_{z \sim p(z)}(g_\theta^*(z)) \right. \\
&\left. \quad + \epsilon \cdot I \|_2]^2 \cdot \| \bigtriangledown_x D[g_\theta^*(z)] + \epsilon \cdot I \|_2^2 \cdot \| J_\theta g_\theta(z) \|_2^2 \right\} \\
&\leq 8(\mathcal{T} + \epsilon)^2 \cdot \mathcal{M}^2 \cdot \epsilon^2.
\end{aligned}
\tag{13}
$$

The proof is done!

### A.2    PROOF OF THEOREM (3.1)

According to eq. (6), we can rewrite the optimization into two parts:

$$
\mathcal{L}(D_c, D_w) = \mathcal{L}_c + \mathcal{L}_w
\tag{14}
$$

where

$$
\begin{aligned}
\mathcal{L}_c = &\min_{D_c, D_w, \|D_w\|_L \leq 1} \mathbb{E}_{(x,y) \sim p_r(x,y)} \| D_c(x) - y \|_2^2 \\
&+ E_{x \sim p_r(x), z \sim p_g(z)} \left\{ D_c(x) - D_c(G(z)) \right\}
\end{aligned}
\tag{15}
$$

and

$$
\mathcal{L}_w = \min_{D_c, D_w, \|D_w\|_L \leq 1} \mathbb{E}_{x \sim p_z} \left\{ D_w(G(z)) \right\} - \mathbb{E}_{x \sim p_r} \left\{ D_w(x) \right\}
\tag{16}
$$

By Eq.(2), we know the optimization of $L_c$ is equivalent to optimizing Eq. (3) in the Improved GAN and we only need to optimize $L_w$. By Theorem 3 in W-GAN (Arjovsky & Bottou, 2017), we know that

$$
\mathbb{E}_{z \sim p(z)}[\nabla_\theta D_w(g_\theta(z))] = \nabla_\theta W(\mathbb{P}_r, \mathbb{P}_\theta)
\tag{17}
$$

By applying Theorem 2.1, we know the first term of $\mathcal{G}$ goes to zero and the second term is given by $\nabla W(\mathbb{P}_r, \mathbb{P}_g)$ Then we derive the desired result. The proof is done.

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
