# OpenReview forum: "Improve Training Stability of Semi-supervised Generative Adversarial Networks with Collaborative Training"
_ICLR.cc/2018/Conference — Reject_

### Official Review · AnonReviewer2 · 2017-11-25
**Claims to address the instability of the "Improved GANs" but does not provide any convincing evidence**

**Rating:** 3
**Confidence:** 4

**Review:**

* Summary *
The paper addresses the instability of GAN training. More precisely, the authors aim at improving the stability of the semi-supervised version of GANs presented in [1] (IGAN for short) . The paper presents a novel architecture for training adversarial networks in a semi-supervised settings (Algorithm 1). It further presents two theoretical results --- one (Theorem 2.1) showing that the generator's gradient vanish for IGAN, and the second (Theorem 3.1) showing that the proposed algorithm does not suffer this behaviour. Finally, experiments are provided (for MNIST and CIFAR10), which are meant to support empirically the claimed improved stability of the proposed method compared to the previous GAN implementations (including IGAN).

I need to say the paper is poorly written and not properly polished. Among many other things:

(1) It refers to non-existent results in other papers. Eq 2 is said to follow [1], meanwhile the objectives are totally different: the current paper seems to use the l2 loss, while Salimans et al. use the cross-entropy;

(2) Does not introduce notations in statements of theorems ($J_\theta$ in Theorem 2.1?) and provides unreadable proofs in appendix (proof of Theorem 2.1 is a sequence of inequalities involving the undefined notations with no explanations). In short, it is very hard to asses whether the proposed theoretical results are valid;

(3) Does not motivate, discuss, or comment the architecture of the proposed method at all (see Section 3).

Finally, in the experimental section it is unclear how exactly the authors measure the stability of training. The authors write "unexpectedly high error rates and poor generate image quality" (page 5), however, these things sounds very subjective and the authors never introduce a concrete metric. The authors only report "0 fails", "one or two out of 10 runs fail" etc. Moreover, for CIFAR10 it seems the authors make conclusions based only on 3 independent runs (page 6).

[1] Salimans et al, Improved Techniques for Training GANs, 2016

---

### Official Review · AnonReviewer3 · 2017-11-27
**Review for "Improve Training Stability of Semi-supervised Generative Adversarial Networks with Collaborative Training"**

**Rating:** 2
**Confidence:** 4

**Review:**

Summary of paper and review:

The paper presents the instability issue of training GANs for semi-supervised learning. Then, they propose to essentially utilize a wgan for semi-supervised learning.

The novelty of the paper is minor, since similar approaches have been done before. The analysis is poor, the text seems to contain mistakes, and the results don't seem to indicate any advantage or promise of the proposed algorithm.

Detailed comments:

- Unless I'm grossly mistaken the loss function (2) is clearly wrong. There is a cross-entropy term used by Salimans et al. clearly missing.

- As well, if equation (4) is referring to feature matching, the expectation should be inside the norm and not outside (this amounts to matching random specific random fake examples to specific random real examples, an imbalanced form of MMD).

- Theorem 2.1 is an almost literal rewrite of Theorem 2.4 of [1], without proper attribution. Furthermore, Theorem 2.1 is not sufficient to demonstrate existence of this issues. This is why [1] provides an extensive batch of targeted experiments to verify this assumptions. Analogous experiments are clearly missing. A detailed analysis of these assumptions and its implications are missing.

- In section 3, the authors propose a minor variation of the Improved GAN approach by using a wgan on the unsupervised part of the loss. Remarkably similar algorithms (where the two discriminators are two separate heads) to this have been done before (see for example, [2], but other approaches exist after that, see for examples papers citing [2]).

- Theorem 3.1 is a trivial consequence of Theorem 3 from WGAN.

- The experiments leave much to be desired. It is widely known that MNIST is a bad benchmark at this point, and that no signal can be established from a minor success in this dataset. Furthermore, the results in CIFAR don't seem to bring any advantage, considering the .1% difference in accuracy is 1/100 of chance in this dataset.

[1]: Arjovsky & Bottou, Towards Principled Methods for Training Generative Adversarial Networks, ICLR 2017
[2]: Mroueh & Sercu, Goel, McGan: Mean and Covariance Feature Matching GAN, ICML 2017

---

### Official Review · AnonReviewer1 · 2017-11-27

**Rating:** 3
**Confidence:** 5

**Review:**

In the paper, the author tried to address the training issue of SSL-GANs. Arguing that the main problem is the gradients vanishing, it proposed a co-training framework which combining the Wasserstein GAN training. The experiments were executed on MNIST and CIFAR-10.

I think the paper made two strong claims, which are not reasonable for me: firstly, it argued that this is the first work to address training issue of SSL-GANs. Actually, the Fisher GAN paper [Youssef et al., 2017] proposed the "New Parametrization of the Critic" for SSL and showed it was very stable. In [Abhishek at al., 2017], the author also addressed how to make the SSL-GANs stable, following the improved GANs paper idea. Secondly, it made an impression that the author thought the main issue of SSL-GANs is the gradient vanishing. Following the paper [Zihang et al., 2017], it is hard to make claim like this.

The co-training framework is not so novel for me, which combined the Wasserstein loss and general GAN loss. Meanwhile, the experimental results are not solid. The baselines listed are not the state-of-the-art. I suggested that the author should compare with some very recent ones, such as [Youssef et al., 2017], [Zihang et al., 2017], [Abhishek et al., 2017], [Jeff et al., 2016].

---

### Decision · Program_Chairs · 2018-01-29
**ICLR 2018 Conference Acceptance Decision**

**Decision:**

Reject

**Comment:**

The paper aims to combine Wasserstein GAN with Improved GAN framework for semi-supervised learning.

The reviewers unanimously agree that:
 - the paper lacks novelty and such approaches have been tried before.
 - the approach does not make sufficient gains over the baselines and stronger baselines are missing.
 - the paper is not well written and experimental results are not satisfactory.